# Leveraging Anthropometric Measurements to Improve Human Mesh Estimation and Ensure Consistent Body Shapes

## Abstract

The basic body shape of a person does not change within a single video. However, most SOTA human mesh estimation (HME) models output a slightly different body shape for each video frame, which results in inconsistent body shapes for the same person. In contrast, we leverage anthropometric measurements like tailors are already obtaining from humans for centuries. We create a model called A2B that converts such anthropometric measurements to body shape parameters of human mesh models. Moreover, we find that finetuned SOTA 3D human pose estimation (HPE) models outperform HME models regarding the precision of the estimated keypoints. We show that applying inverse kinematics (IK) to the results of such a 3D HPE model and combining the resulting body pose with the A2B body shape leads to superior and consistent human meshes for challenging datasets like ASPset or fit3D, where we can lower the MPJPE by over 30 mm compared to SOTA HME models. Further, replacing HME models estimates of the body shape parameters with A2B model results not only increases the performance of these HME models, but also leads to consistent body shapes.

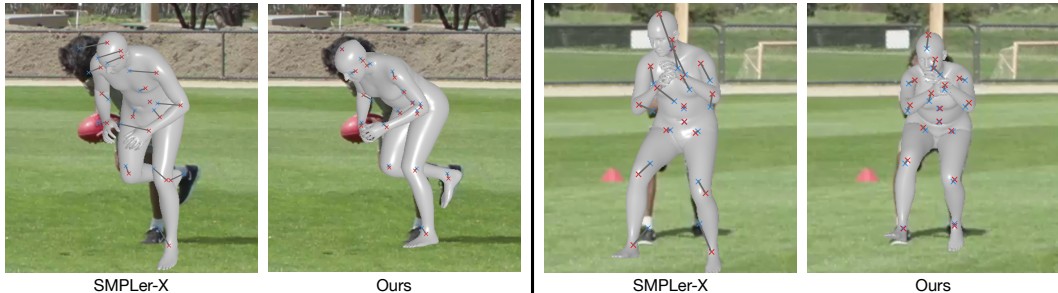

Figure 1: Two qualitative examples from the ASPSet sports dataset. The result from a SOTA HME model, SMPLer-X (Cai et al., 2024), is shown on the left, the result from our model on the right, respectively. GT joints and estimated joints are color-coded. Corresponding joints are connected.

## 1 Introduction

Creating an accurate 3D mesh from monocular images or videos creates new opportunities in fields like 3D animation, gaming, fashion, sports, etc. In many of these application fields, videos are of main interest. While applying HME to videos, one would expect to get a mesh sequence per person with those meshes just changing their poses, but not their body shape. However, analyses of results of SOTA HME models in challenging scenarios show that the body shape of the meshes of the same person differs from frame to frame. This is because most HME models work per image and are unable to process a whole video. Worse, an analysis of currently used 3D mesh and pose datasets reveals the same inconsistencies in the provided ground truth (GT) data. For a precise body posture analysis, as it is necessary in many sports disciplines, an exact model of the athlete's body shape is required. Therefore, many professional athletes are measured anthropometrically during performance assessments. Moreover, the body shape of an actor performing motions for 3D

animations needs to be consistent as the basic body shapes does not change during performances. Changes are only due to different poses and are modeled separately besides the core shape. Thus, the changing body shapes of HME models for the same person are highly unwanted and simply wrong.

Our work aims to create a single perfectly fitting basic body shape for each human and reuse it for all videos with this person. Measuring the human body has already been done for centuries to fit suits or dresses perfectly to a specific body shape. Athletes are also measured for decades for precise performance assessments. In many applications, measuring the person in action would add only a marginal overhead, but ease the postprocessing steps. This is the reason why we propose to use these measurements. However, body shape parameters of common human mesh models like SMPL-X (Pavlakos et al., 2019) are not human interpretable. Therefore, it is not possible to obtain the perfect body shape parameters by anthropometric measurements. Hence, we train a machine learning model (called *A2B*, anthropometric measurements to body shape) to translate those measurements into body shape parameters for HME. With this approach, measuring a person once creates the body shape that can be used for all frames in all evaluation videos.

HME models are currently performing well on everyday data. However, in more challenging scenarios like sports, their performance is still inferior to finetuned SOTA 3D HPE models. 3D HPE models only predict 3D keypoints resulting in a stickfigure pose, whereas HME models output a posed mesh including the human's surface. Due to the lack of GT meshes, HME models cannot be trained on datasets with solely 3D keypoint annotations which are easier to create than mesh annotations. The usage of synthetic data is emerging in the field, but is not applicable to challenging or specific scenarios like sports. In this paper, we propose a solution to that problem. With our A2B model and anthropometric measurements, we can now create the body shape parameters of humans needed for HME. We further apply inverse kinematics (IK) to produce the rotations that are missing in the 3D stick-figure model that is created by 3D HPE models. Together with our A2B body shape, we are able to generate human meshes that have a consistent body shape and a precise pose. Evaluations show that the performance of this composed approach is superior compared to HME models. We show qualitative examples of our model and a SOTA HME model, SMPLer-X (Cai et al., 2024), in Figure 1. Our approach is generally applicable to any HME problem. We choose sports datasets to validate our proposed approach, since the poses in sports change rapidly, which exacerbates the problem of inconsistent body shapes. Our contributions can be summarized as follows:

- We reveal inconsistencies in the GT data of ASPset (Nibali et al., 2021) and fit3D (Fieraru et al., 2021). The body shape of a single person varies mistakenly in the GT.
- We create and evaluate different models for converting between anthropometric measurements and SMPL-X body shape parameters. We call these models *A2B*. We are the first to intoduce a possibility to convert from anthropometric measurements to body shape parameters.
- We analyze and compare the performance of existing HME models on ASPset and fit3D. Replacing the estimated body shape parameters (and keeping the pose) of each HME model with A2B body shape parameters increases the performance of all models.
- With finetuned SOTA 2D and 3D HPE models (Xu et al., 2022; Einfalt et al., 2023), inverse kinematics (IK), anthropometric measurements, and our A2B model, we estimate accurate human meshes with a consistent body shape. We show that this approach achieves superior results to SOTA HME models, although still evaluated on the inconsistent GT.
- Our models and code for our approach are publicly available: https://anonymous.4open.science/r/A2B_human_mesh-FA54/

## 2 RELATED WORK

Human Mesh Estimation (HME) is an active area of research. Body models like SMPL (Loper et al., 2015) and its successor SMPL-X (Pavlakos et al., 2019) are broadly used. Their advantage is that they decouple human pose and shape. The pose parameters $\theta$ give the rotations of the joints relative to the parent joint. The shape parameters $\beta$ model the basic body shape. At first, a mesh is created with a linear mapping from $\beta$ parameters to a T-shaped pose. Next, some pose-specific shape deformations are applied and then the mesh is rotated at the joints according to the $\theta$ parameters.

The first HME model that estimates SMPL-X meshes from images was introduced along SMPL-X (Pavlakos et al., 2019). This model, called SMPLify-X, detects 2D image features and then fits an

SMPL-X model to these. To achieve that, they incorporate a pose prior trained on a large motion capture dataset and an interpenetration test. A more recent model for HME is Multi-HMR (Baradel et al., 2024). It predicts 2D heatmaps for person centers and based on that the human mesh with a human prediction head. OSX (Lin et al., 2023) is another HME model. It uses a component aware transformer that is composed of a global body encoder and local decoders for face and hands. SMPLer-X (Cai et al., 2024) is introduced as a generalist foundation model for HME, which is trained on a large amount of datasets. It mainly uses vision transformers. There is a multitude of other recent HME models, some focussing more on whole-body HME (Choutas et al., 2020; Feng et al., 2021; Moon et al., 2022), others on multi-person HME - either with a two stage approach using a person detector and a single person human mesh estimator (Choi et al., 2022; Goel et al., 2023; Qiu et al., 2022), or a single stage approach estimating the meshes of all persons at once (Qiu et al., 2023; Sun et al., 2021; Zhang et al., 2021).

Choutas et al. (2022) observed that existing HME models focus more on the body pose than the shape, although the shape is equally important for many applications. They propose SHAPY, a model that uses anthropometric and linguistic attributes to create accurate body shapes. Moreover, Sarkar et al. (2023) introduce SoY, which contains specific loss functions to enhance the body shape accuracy. They further propose a refinement step during test time that enhances the shape quality even more. AnthroNet (Picetti et al., 2023) propose a new body model that is learned with an end-to-end trainable pipeline. It takes anthropometric measurements as an input to learn a mesh model that accurately captures shapes of humans, but this model is different from the commonly used SMPL-X model. We use the common SMPL-X body model and decouple the estimation of the shape from the estimation of the pose. This way, we ensure the consistency of the body shape over time.

Inverse kinematics (IK) are common in the field of robotics. In the last years, 3D HME approaches leveraged IK to enhance their output. HybrIK (Li et al., 2021) transforms 3D joint coordinates to relative body-part rotations for 3D HME by using a twist-and-swing decomposition. HybrIK-X (Li et al., 2023) further enhances HybrIK with expressive face and hands. Cha et al. (Cha et al., 2022) leverage IK to tackle the challenge of person-to-person occlusions in images with interacting persons. PLIKS (Shetty et al., 2023) (Pseudo-Linear Inverse Kinematic Solver) approaches HME as a model-in-the-loop optimization problem by analytically reconstructing the human model via 2D pixel-aligned vertices in an IK-like manner.

Although HME is an active area of research, it is yet not common in computer vision for sports. Due to high velocities and a great variation of poses, sports is a challenging scenario for all kinds of human pose and shape estimation. The fit3D dataset (Fieraru et al., 2021) is a dataset which consists of videos from gym sports exercises with repetitions and is annotated with human meshes. AIFit (Fieraru et al., 2021) is a tool trained on fit3D which can reconstruct 3D human poses, reliably segment exercise repetitions, and identify the deviations between standards learned from trainers, and the execution of a trainee to give feedback to trainees. Other sports datasets only consist of 3D joint annotations, like ASPset (Nibali et al., 2021) or SportsPose (Ingwersen et al., 2023). SportsCap (Chen et al., 2021) is an approach for simultaneously capturing 3D human motions and understanding fine-grained actions from monocular challenging sports videos.

## 3 ERRORS IN 3D HUMAN SHAPE GROUND TRUTH

Each person has a specific body shape that does not change over a short time period. Therefore, the SMPL-X body model decouples the human pose encoded by $\theta$ parameters from the body shape encoded by $\beta$ parameters. Deformations to the mesh that are caused by the pose are modeled separately. Therefore, it makes sense to assign a single set of shape parameters $\beta$ to a person to describe his/her shape for a given short time period such as a recorded action. Further, there are lengths that can be calculated from 3D joints that should never change, since individual bones of humans are rigid and should not be deformed by different poses. Our approach enforces a single set of shape parameters per person and immutable bone lengths.

As a first step, we analyze if the GT data of our used datasets fulfills these properties. In this paper, we use ASPset (Nibali et al., 2021) and fit3D (Fieraru et al., 2021), since both datasets consist of videos with fast changing poses and 3D GT. Results for the Human3.6m (Ionescu et al., 2014) and MPI-INF-3DHP (Mehta et al., 2017) datasets are presented in the supplementary. For ASPset, we analyze bone lengths, since it has only GT annotations for 3D joints. For fit3D, GT SMPL-X $\beta$

Table 1: GT data analysis for ASPset (left) and fit3D (right): Standard deviation $\sigma$, relative standard deviation $\frac{\sigma}{avg}$ and relative range $\frac{\max - \min}{avg}$ of anthropometric measurements. Standard deviations are given in cm, but for the $\beta$ parameters. The values are averaged between left and right body parts and between all persons in the test set of each dataset. The $\beta$ parameter standard deviation is averaged over all $\beta$ parameters.

| | ASPset | | | fit3D | | | |
| Measurement | $\sigma$ | r. $\sigma$ | r. range | Measurement | $\sigma$ | r. $\sigma$ | r. range |
|---|---|---|---|---|---|---|---|
| head | 0.91 | 5.98% | 57.91% | head | 0.69 | 2.60% | 14.70% |
| hip width | 1.71 | 9.48% | 85.46% | hip circ. | 0.80 | 0.76% | 5.44 % |
| forearm | 1.99 | 8.37% | 92.04% | forearm | 0.33 | 1.40% | 7.64% |
| upper arm | 1.72 | 6.29% | 66.35% | arm | 0.76 | 1.53% | 8.32% |
| lower leg | 1.44 | 3.60% | 41.36% | lower leg | 0.43 | 1.14% | 11.14% |
| thigh | 1.65 | 4.23 % | 35.46% | thigh | 0.37 | 1.03% | 9.21% |
| | | | | height | 1.65 | 0.98% | 6.45% |
| | | | | $\beta$ param. | 0.65 | | |

parameters are available, hence we can analyze the $\beta$ parameters directly and further the derived anthropometric measurements. These values are the output of our deterministic B2A function: It generates a standard T-pose with the given $\beta$ parameters and computes 36 anthropometric measurements from the resulting mesh. Results of our GT analysis for a subset of the anthropometric values are shown in Table 1. We can see that the GT itself is not consistent. The deviations are larger for ASPset, but although we have GT SMPL-X meshes for fit3D, the $\beta$ parameters of a single person have a standard deviation of 0.65 on average.[1] This is a relevant flaw in the GT shape annotation, since based on the model, the GT shape should be static for each human. Nevertheless, we use the given inconsistent GT for our evaluations for comparability with related work and as we have no good means to correct them. The reader should keep this in mind. Nevertheless, we want to encourage future research in the field of 3D human pose and mesh data collection to try to eliminate these flaws in the provided GT.

## 4 FROM ANTHROPOMETRIC MEASUREMENTS TO THE BODY SHAPE

Humans have been measured for centuries (Doyon et al., 2023). Tailors know exactly which measurements to take for perfectly fitting a suit or dress to the body shape of a customer. In sports, professional athletes are measured for precise performance assessments, too. Measuring a human is easy and well understood. In contrast, the parameters of the body shape for human mesh models like SMPL-X (Pavlakos et al., 2019) are not humanly interpretable. The $\beta$ parameters describe the principal components of the human body shape with typically around 10 to 16 values and are the result of a PCA executed on the human meshes of a training dataset while learning the SMPL-X model. Fixing all $\beta$ parameters despite one and looking at the results lets human observers get a notion of what this parameter might mean, but in total, the $\beta$ parameters and their interactions are not well interpretable. Therefore, we want to leverage the well established technique of measuring humans to create precise body shape parameters for the commonly used SMPL-X human mesh model. We call our approach to convert from 36 Anthropometric measurements to Body shape parameters *A2B*. Since there is no known relation between anthropometric measurements and $\beta$ parameters, our aim is to learn this mapping. **We are the first to present a method to convert from anthropometric measurements to body shape parameters**. The reverse direction, *B2A*, is a deterministic function of the human mesh, as the anthropometric measurements can be measured from the mesh.

### 4.1 DATA GENERATION

We select the 36 anthropometric measurements for our models based on the selections of AnthroNet (Picetti et al., 2023) and an anthropometry study of the U.S. army (Gordon et al., 2014). They can be categorized into 23 lengths and 13 circumferences. Apart from the bone lenghts like arm length, thigh length, etc., this includes also detailed measurements like shoulder width, front torso

---

[1] Averaged standard deviation means (in the whole paper) that the standard deviation is calculated per person, and the mean of the resulting standard deviations is calculated afterwards.

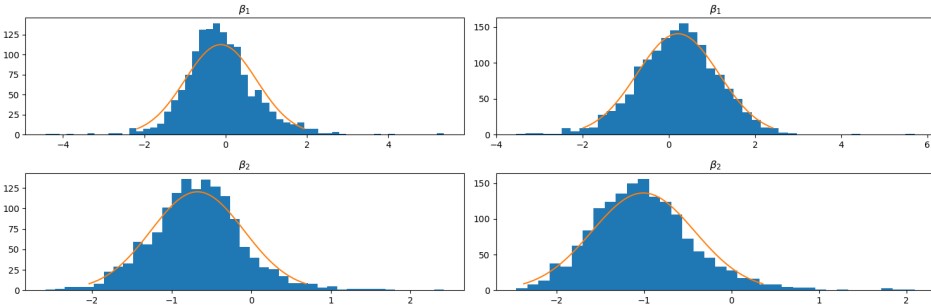

Figure 2: Histograms and fitted normal distribution (orange) for the first two $\beta$ parameters for all male (left) and female (right) subjects of the AGORA (Patel et al., 2021) dataset.

height, lateral neck length, waist circumference, calf circumference, etc. A visualization and precise description of the selected measurements can be found in the supplementary.

Many existing datasets provide a wide range of different poses, but most incorporate the same humans. In order to learn a conversion model from anthropometric measurements to $\beta$ parameters, we need a lot of samples for different humans, no matter the pose. With given shape parameters, we can use the B2A function to compute the anthropometric measurements. Recall, B2A is a deterministic function measuring the anthropometric values from the meshes.

Because of the need for many different body shapes for the learning process, we use the AGORA (Patel et al., 2021) dataset. It consists of 1447 male and 1588 female subjects, which is comparably large. We are not able to use the larger dataset from AnthroNet (Picetti et al., 2023), since it uses its own mesh model and the authors did not publish their conversion to the SMPL-X model, which we want to use as it is most commonly used in research. Although comparably large, 1447/1588 subjects is still a very little amount of data to learn a model. Hence, we analyze the $\beta$ parameters in the AGORA dataset with the aim to randomly sample more data with realistic body shapes. Histograms (see Figure 2) of the occurring $\beta$ parameters show that their distribution roughly follows a normal distribution. Therefore, we train our models with randomly sampled data according to the distributions, either assuming a normal distribution fitted to the histograms or a uniform distribution with the same minimum and maximum values as in the data analysis. This means that we sample each $\beta$ parameter according to the selected distribution, create the mesh according to the sampled values and derive the anthropometric measurements with B2A. With this strategy, we can create a dataset with as many subjects as we need. We expect the analyzed data to not cover the full range of human body shapes as AGORA is a synthetic dataset. Therefore, we also train with extended distributions, meaning that we increase the standard deviation $\sigma$ to $\alpha_n\sigma$ in the case of a normal distribution or stretch the interval by a factor $\alpha_u$ in case of a uniform distribution.

### 4.2 MODELS

We use the same number of $\beta$ parameters for each gender as used in the AGORA dataset, meaning 11 for male, 10 for female, and 16 for neutral subjects. With 36 anthropometric measurements as input values and $10 - 16$ output values for our A2B models, the dimensionality of the data is low. Therefore, we experiment with Support Vector Regression (SVR) and with small neural networks (NN). We split the AGORA dataset in a 80% train, 15% test and 5% validation subset. For SVR, we additionally randomly sample $10,000$ subjects for training. We use a hyperparameter search based on the validation split to determine the optimal settings, which leads us to a radial basis function kernel, an error margin of $\epsilon = 0.012$ and a regularization constant of $C = 3791$. For the NNs, we randomly sample new data in each iteration. The hyperparameter search for the NNs results in a model with 4 layers, 330 neurons per layer, tanh as activation function, and Xavier Gloriot as initialization. We use mean squared error on the model output (the $\beta$ parameters) to train the models.

### 4.3 RESULTS

We train each model (NN and SVR) for each gender and with different dataset variants: We train solely on the AGORA train split, as well as on uniformly and normally distributed randomly sampled data according to the data analysis, and we further extend the range of the data as described in

Table 2: Results of our A2B models on the test split of the AGORA dataset. The first block ($\beta$) shows the error if we take the GT $\beta$ parameters, derive 36 anthropometric measurements (B2A), input them into the A2B models and evaluate the MSE of the predicted $\beta$ parameters in the scale $10^{-3}$. The second block ($A$) calculates B2A from the predicted $\beta$ parameters and evaluates the mean difference between the GT and predicted anthropometric measurements (all 36) in mm. Results are given for m(ale), f(emale), and n(eutral) models. A visualization is provided in the supplementary.

| Model | train data | $\beta$ | | | $A$ | | |
|---|---|---|---|---|---|---|---|
| | | m | f | n | m | f | n |
| NN | AGORA | 9.11 | 13.9 | 24.0 | 0.814 | 0.934 | 1.459 |
| NN | normal | 2.62 | 4.34 | 18.0 | 0.356 | 0.392 | 1.711 |
| NN | normal ext. | 1.87 | 3.69 | **14.8** | 0.248 | 0.285 | **1.384** |
| NN | uniform | 5.08 | **1.25** | 18.3 | **0.243** | **0.268** | 2.381 |
| NN | uniform ext. | **1.61** | 3.20 | 16.3 | 0.274 | 0.419 | 1.774 |
| SVR | AGORA | 2.56 | 16.1 | 3.82 | 1.659 | 5.195 | 2.557 |
| SVR | normal | 4.08 | 17.8 | 59.0 | 2.975 | 4.303 | 14.63 |
| SVR | normal ext. | 0.210 | 4.60 | 6.27 | 0.280 | 1.090 | 2.211 |
| SVR | uniform | 0.0396 | 0.0350 | **0.162** | 0.124 | 0.284 | 0.214 |
| SVR | uniform ext. | **0.0252** | **0.0193** | 0.306 | **0.082** | **0.136** | **0.164** |

Section 4.1 with $\alpha_n = \alpha_u = 1.5$. The results are displayed in Table 2. We evaluate the performance of our models in two ways. At first, we calculate the error of the predicted and GT $\beta$ parameters. Second, we calculate the mean deviation of the anthropometric measurements of the meshes from the predicted and GT $\beta$ parameters ($A$). Therefore, this evaluation can further be seen as a kind of cycle consistency evaluation of A2B (our learned model) and B2A (the deterministic measuring function). We provide a visualization of the evaluation process in the supplementary. The anthropometric error is our main metric as these values reflect the desired body shape given as an input by the user and are further interpretable. The $\beta$ parameters are somehow arbitrary in their scale. For all genders and SVR, using an extended uniformly sampled dataset works best. For the NNs, a uniformly sampled dataset works best for male (m) and female (f) genders and an extended normally sampled dataset for the neutral (n) meshes. The results for the neutral model are worse in general, especially in the case of the NNs, which might be due to the fact that the neutral model needs to express a more diverse range of body shapes. Furthermore, the SVR achieves better results for all genders. Thus, we use these models now for all datasets, without any finetuning or adaptation to specific datasets.

## 5 LEVERAGING A2B MODEL RESULTS FOR HME

Now that we have trained the A2B models, we can use them to generate precise body shape parameters upfront and reuse them for every evaluation of a specific person. In the next section, we describe how the A2B results can be used to improve existing HME models (see Section 5.1). Further, **we introduce a new approach to HME** (see Section 5.2). We leverage the good performance of a sequence-based 2D-to-3D uplifting HPE model and convert the 3D (stick-figure) poses to human meshes with the help of the A2B models. With this new approach, we achieve superior results compared to existing HME models.

We evaluate all models on ASPset (Nibali et al., 2021), which is a 3D human pose dataset. It consists of various different sports motion clips performed by different subjects, recorded from three camera perspectives. We evaluate on the test set, which contains two subjects and 30 videos for each subject. In the test set, only one camera perspective is public, so we evaluate on this perspective. Evaluating SMPL-X meshes for ASPset is non-trivial. Regressing standard SMPL-X joints from SMPL-X meshes is built-in, but for all other keypoint definitions it is necessary to define a custom regressor. Since there is no regressor available for ASPset, we create a custom SMPL-X regressor for the keypoints head, head top, neck, l./r. hip, and torso. For left ankle and elbow, we mirror the corresponding right regressor, as it is not the exact mirrored version in the standard regressor. We use the code from (Russo, 2020) and update it to our needs.

We further evaluate on fit3D (Fieraru et al., 2021), since this is the only sports dataset with public SMPL-X annotations. We evaluate on the SMPL-X joints here since they are available. We select a

subset of 37 SMPL-X joints. Since our focus is mainly on the body and not on the hands and face, we remove a lot of these joints. A list of the selected joints can be found in the supplementary. For both datasets, we do not have access to the athletes to measure their anthropometric values. Therefore, we use the ground truth to obtain the anthropometric measurements that are needed as an input for the A2B models, details can be found in the supplementary. Since there is no GT available for the official test set evaluation on the evaluation server of fit3d, we split the official training dataset into a training, validation, and test set for our evaluations. We choose the last subject (s11) with all 188 videos as the test set.

Sports datasets differ from most commonly used everyday activity datasets in the aspect that the poses are more diverse and the motions are faster, which makes sports datasets more difficult. In some cases, the poses are so difficult that some models do not detect a human at all. This makes a fair evaluation hard, since the standard MPJPE metric takes the mean of the joint position errors. Assuming a default pose for all frames where no person is detected would result in a very high error that shifts the mean enormously. Hence, we report the MPJPE only on the frames where persons are detected. Since mostly difficult frames are omitted, this will result in a slightly easier setting for methods that find less persons, but we include the number of missing frames in our results for comparison.

## 5.1 IMPROVING HME MODEL RESULTS

As already described, a major problem for HME based analyses is a varying body shape within a single video. Using $\beta$ parameters generated with A2B models solves this problem. The necessary 36 anthropometric measurements are either measured from the human directly or - as in our case - averaged from the provided GT (fit3D) or averaged from IK applied to the GT poses (ASPset). We call these measurements pseudo GT in the following. More details can be found in the supplementary. We combine existing HME models with the body shape estimated by our A2B models by replacing the estimated $\beta$ parameters with the ones predicted by the A2B models. We select three recent well performing models on the AGORA dataset (SMPLer-X (Cai et al., 2024), OSX (Lin et al., 2023), Multi-HMR (Baradel et al., 2024)), and the first HME method developed by the SMPL-X authors, SMPLify (Pavlakos et al., 2019). Since SMPLer-X is trained on the official training data of fit3D, an evaluation with this model is not meaningful and we omit it. Moreover, SMPLify-X is not SOTA any more and achieved the worst results for ASPset, therefore we omit it, too.

The first evaluation contains the original result from the respective model, and evaluations where the pose from the model is kept, but the $\beta$ parameters are replaced with the A2B body shape parameters with pseudo GT input. In order to emphasize the variation of the body shape in the HME model results, we evaluate the standard deviation of the body height per person. Results are displayed in Table 3. We can see that for all models, replacing the estimated $\beta$ parameters by $\beta$ parameters from our A2B models with pseudo GT input leads to an improvement. For some models, the male meshes outperform the neutral ones and vice versa. The test sets of both datasets contain only male subjects, therefore we cannot evaluate female meshes here. Interestingly, the NN outperforms the SVR for all neutral experiments, although the SVRs achieved better results on the AGORA dataset evaluation. The reason could be that AGORA is a synthetic dataset and does not reflect reality. SMPLer-X achieves the best results for ASPset and Multi-HMR for fit3D, both with a significant margin. OSX performs worse on fit3D than on ASPset, but Multi-HMR performs better by a large margin and surpasses OSX. All methods benefit from our A2B $\beta$ parameters based on pseudo GT with improvements from 11 mm to 2 mm.

We further evaluate the capabilities of a fixed body shape without available GT anthropometric measurements to ensure consistent body shapes also in the case that no measurements are available. Hence, we use the meshes originally created by the HME models, compute the anthropometric measurements of each mesh with B2A, calculate their median values and convert them to $\beta$ parameters via the A2B models. Results are displayed in Table 4. For SMPLer-X and OSX, using the median $\beta$ parameters lead to worse results on ASPset. Regarding ASPset, for all models except Multi-HMR, using our A2B models increases the performance slightly. Switching from the neutral output that these models all have to a gendered model works best in these cases, but the neutral A2B models also lead to a marginal improvement. Regarding fit3D, using the median $\beta$ parameters already enhances the results. Using $\beta$ parameters from an A2B model leads to further improvement, OSX achieves the best results with SVR and the male model, Multi-HMR with SVR and the neutral model.

Table 3: MPJPE results in mm for existing models on the test splits of ASPset (top) and fit3D (bottom). The second column (*orig.*) contains the original results, the other columns results with replaced $\beta$ parameters from our **A2B models with pseudo GT anthropometric measurements** as input and either male (m) or neutral (n) meshes, and the percentage of frames with no result (no r.). The $\sigma$ column displays the mean standard deviation of the body height per subject in cm for the original results. With A2B body shapes, $\sigma = 0$.

| | Model | orig. | $\sigma$ | NN m | SVR m | NN n | SVR n | no r. ($\downarrow$) |
|---|---|---|---|---|---|---|---|---|
| ASPset | SMPLer-X (Cai et al., 2024) | 86.02 | 1.8 | 78.88 | 78.50 | **78.38** | 78.47 | 0.11% |
| | OSX (Lin et al., 2023) | 92.34 | 0.2 | 89.58 | **89.28** | 89.44 | 89.56 | 0.10% |
| | Multi-HMR (Baradel et al., 2024) | 102.54 | 3.6 | 100.04 | 100.25 | **99.28** | 99.53 | 0.44% |
| | SMPLify-X (Pavlakos et al., 2019) | 138.18 | 13.0 | 127.73 | 127.36 | **126.78** | 126.89 | 0.02% |
| fit3D | OSX (Lin et al., 2023) | 99.20 | 4.8 | 93.22 | 92.58 | **91.72** | 92.21 | 3.45% |
| | Multi-HMR (Baradel et al., 2024) | 72.87 | 3.0 | 70.72 | 71.65 | **70.70** | 71.74 | 1.54% |

Table 4: MPJPE results in mm for existing models on the test split of the ASPset (top) and fit3D (bottom) datasets. The second column contains the original results, the other columns results with replaced $\beta$ parameters. Either the median $\beta$ parameters are used or the results from our **A2B models with median anthropometric measurements from the respective model** as input. For *no r.* see Table 3.

| | Model | orig. | median | NN m | SVR m | NN n | SVR n |
|---|---|---|---|---|---|---|---|
| ASPset | SMPLer-X (Cai et al., 2024) | 86.02 | 86.04 | 85.89 | **85.69** | 86.03 | 85.99 |
| | OSX (Lin et al., 2023) | 92.34 | 92.37 | 92.44 | **92.17** | 92.33 | 92.38 |
| | Multi-HMR (Baradel et al., 2024) | 102.54 | **102.05** | 102.59 | 102.96 | 102.07 | 102.16 |
| | SMPLify-X (Pavlakos et al., 2019) | 138.18 | 133.59 | 133.81 | **133.46** | 133.56 | 133.48 |
| fit3D | OSX (Lin et al., 2023) | 99.20 | 96.70 | 98.20 | **96.07** | 96.60 | 96.72 |
| | Multi-HMR (Baradel et al., 2024) | 72.87 | 72.08 | 72.62 | 72.37 | 72.06 | **71.92** |

They can further be used to easily convert between neutral and gendered (male or female) models. In contrast, $\beta$ parameters are not transferable between models of different genders. Therefore, until now, the conversion could only be achieved by minimizing the vertex error between meshes of different genders in an iterative process. With our approach, we can now use the B2A fuction to obtain anthropometric measurements for a mesh of one gender and apply the A2B model of the other gender to these anthropometric measurements in order to get the corresponding $\beta$ parameters for this gender.

## 5.2 HUMAN MESH ESTMATION WITH SEQUENCE BASED 3D HPE AND A2B RESULTS

All evaluated HME models are working image-wise. In contrast, SOTA 3D HPE models take a long sequence of 2D poses as an input, which helps to capture movements precisely. The models are called uplifting models, since they lift 2D pose sequences to 3D pose sequences. We use the effcent SOTA 3D HPE model *uplift and upsample* (UU) (Einfalt et al., 2023) to estimate the 3D poses on videos. To estimate the required 2D poses from the video frames, we use ViTPose (Xu et al., 2022), a SOTA 2D pose estimation model. It is important to note that UU operates on pose sequences instead of single frames like the HME models in Section 5.1 and can leverage the information of neighboring frames to estimate a more sophisticated pose. Since we have GT 3D joints available, we can finetune the models (ViTpose for 2D HPE and UU for 3D HPE) on our data. This is also necessary to adapt the model to the dataset specific joint definitions since many 3D HPE models like UU are pretrained on datasets like Human3.6m (Ionescu et al., 2014), but those joint definitions do not match ASPset nor fit3D. We finetune both 2D and 3D HPE models on the training subsets. On the test subsets, UU achieves an MPJPE of 63.85 mm on ASPset and an MPJPE of 29.60 mm on fit3D, which is better than the best existing HME model for both datasets (see Section 5.1). However, UU only outputs 3D joints, no meshes. Moreover, a stick-figure 3D pose is not sufficient to model the pose parameters $\theta$ of the SMPL-X mesh, since some rotations are missing. Hence, it is impossible to calculate the necessary rotation parameters directly from the UU result.

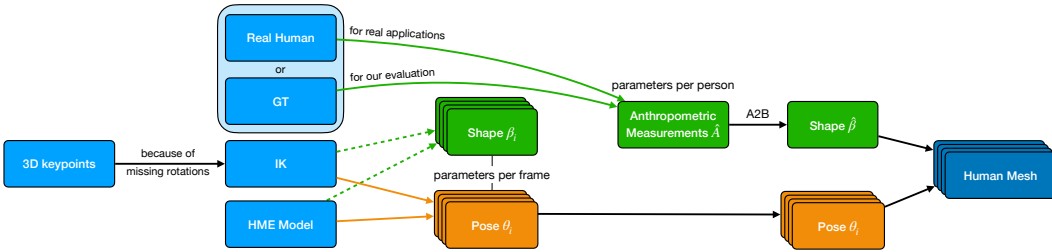

Figure 3: Overview of our approach. The pose and shape parameters are obtained either from IK applied to UU results (Sec. 5.2.2) or from an HME model (Sec. 5.1). In real applications, the anthropometric measurements will be taken directly from the humans . For our evaluations, we use the GT shape parameters and further experiment with the shape parameters of the respective model (IK or HME).

### 5.2.1 INVERSE KINEMATICS FOR FULL POSE ESTIMATION

Therefore, we use the well established approach of inverse kinematics (IK) to obtain the missing rotations by fitting an SMPL-X mesh to the 3D joint locations estimated by UU. We use the inverse kinematics approach with the VPoser extension (Pavlakos et al., 2019). VPoser is a learned prior for human poses, since the raw SMPL-X model definition allows impossible poses for humans. VPoser learned plausible poses from the large AMASS (Mahmood et al., 2019) dataset and helps IK to generate only plausible poses. IK learns the best SMPL-X parameters ($\beta$ and $\theta$) that fit the mesh to the given 3D joint locations by minimizing the error between the given joint locations and the regressed joint locations from the mesh. IK is an iterative algorithm and adjusts the pose and the shape parameters with a gradient descent minimization approach in each step. Besides the already described joint error, IK further penalizes abnormal poses with a VPoser error and extreme body shapes with a $\beta$ parameter error. We execute IK per frame, which results in a slight jitter in between the frames, but leads to more accurate joint positions. Since IK needs multiple iterations to adjust the standard T-pose parameters to achieve a pose that is roughly close to the desired UU pose, we speed up the process by initializing the pose and shape parameters with the result from the previous frame if available. This also enhances the final result slightly.

### 5.2.2 EXPERIMENTS AND RESULTS

We evaluate different experiments in Table 5. For comparison, we mention the UU 3D HPE performance (first rows for each dataset in Tab. 5). These results correspond to stick-figure poses and not the required human meshes. Therefore, they are not directly comparable to the other results.

Our main approach is displayed in the second rows in Table 5, respectively. We evaluate the results of IK applied to the UU joint locations with original, median and pseudo GT based A2B $\beta$ parameters. The final real-world scenario corresponds to the following approach. GT anthropometric values can be measured from the athlete directly and the 3D pose can be estimated with UU and IK. The $\beta$ parameters are estimated with the A2B models. A visualization of this pipeline can be found in Figure 3. We include the best result from existing HME models in the respective last rows for comparison and provide qualitative results in Figure 1. **Our approach outperforms the best existing HME model for both datasets by a large margin.**

We analyze the results of the building blocks of our model in detail. Applying IK to the UU results deteriorates the UU results by nearly 4 mm for ASPset and 8 mm for fit3D (see Tab. 5, first and second rows, column *orig.*), but the UU result is only a stick-figure pose and not a human mesh, hence it is not sufficient for our purpose. Moreover, these results are still better than the best existing HME model.

Next, we replace the inconsistent $\beta$ parameters with the results from our A2B models. For ASPSet, using pseudo GT anthropometric measurements results in a large improvement of over 12 mm. Remarkably, this result surpasses even the original UU result by 8 mm. It seems that incorporating a clearly defined mesh helps to fix some typical errors of UU and enhance its result in case of ASPset. In general, the error on fit3D is much lower for UU based approaches. The reason might be that it

Table 5: MPJPE results in mm on the test splits of ASPset (top) and fit3D (bottom) of our approach compared to the respective best HME model. The *pose* column indicates the origin of the pose. The *orig* column contains the result as it is estimated from the method indicated in the *pose* column (with inconsistent body shapes). The right block contains the results with the originally estimated $\beta$ parameters replaced by consistent ones. The *measurements* column indicates which anthropometric measurements are used for the A2B computation (which $beta$ parameters are used for the median computation) whose results are the replacement $\beta$ parameters in the last five columns. We highlight the best results for *estimated* meshes with *consistent shapes*.

| DS | inconsistent shape | | consistent shape (ours) | | | | | |
| | pose | orig. | measurements | NN m | SVR m | NN n | SVR n | median |
|---|---|---|---|---|---|---|---|---|
| ASPset | UU | 63.85 | no mesh | | | | | |
| | IK-UU | 67.54 | GT | 56.44 | 56.56 | **55.14** | 55.19 | - |
| | IK-UU | 67.54 | IK-UU | 66.92 | 66.60 | 67.25 | 67.12 | 67.16 |
| | SMPLer-X | 86.02 | GT | 85.89 | 85.69 | 86.03 | 85.99 | 86.04 |
| fit3D | UU | 29.60 | no mesh | | | | | |
| | IK-UU | 36.89 | GT | 39.36 | 38.69 | **37.46** | 37.89 | - |
| | IK-UU | 36.89 | IK-UU | 40.33 | 38.82 | 38.14 | 38.10 | 38.29 |
| | Multi-HMR | 72.87 | GT | 72.62 | 72.37 | 72.06 | 71.92 | 72.08 |

consists of much more data, such that we can finetune UU for a longer time. Further, the videos are recorded in a lab in comparison to the in-the-wild videos of ASPset. Therefore, the results of ASPset are more relevant for future applications of our approach, where we assume only a few available 3D annotations and in-the-wild recordings. For fit3D, applying the A2B body shapes from pseudo GT anthropometric measurements leads to a slight decrease in performance of 0.6mm. Inconsistent shapes in the GT (see Section 3) are likely to cause this behavior. Still, our approach using a 3D HPE model and IK outperforms all existing HME models, no matter if the original inconsistent or the consistent body shapes from A2B are used.

As already described in Section 5.1, we further evaluate the capabilities of a **consistent shape without available GT anthropometric measurements**. The naive approach is to use the median of the estimated inconcistent $\beta$ parameters (see Tab. 5, column *median*). Another approach is to use the meshes created by IK applied to the UU results, compute the anthropometric measurements with B2A, calculate the median anthropometric values and convert them to $\beta$ parameters via the A2B models. Results are displayed in Table 5, row three. For ASPset, using fixed body shape parameters from A2B models based on the measurements from UU results achieves a slightly better score than the results with inconsistent body shapes. For fit3d, the MPJPE increases by another 0.6 mm, but **the A2B model results are a slighly better alternative for consistent body shapes compared to the median $\beta$ parameters**.

Further, our approach with IK can be used to generate pseudo GT meshes for datasets with only 3D keypoint annotations. We can use these pseudo GT meshes to finetune HME models and increase their performance regarding the estimated keypoints for the specific dataset. However, these results are still worse than the results of our approach. We present the results in the supplementary.

## 6 CONCLUSION

In this paper, we address the problem of inconsistent estimated body shapes of humans in videos. We analyze the GT data of 3D pose and mesh datasets and find inconsistencies already in their annotations. Then, we propose a family of learned *A2B* models to convert 36 anthropometric measurements to SMPL-X $\beta$ parameters. This can be used to measure a human once and use the resulting shape of the A2B model for all evaluations. With this strategy, the body shape is accurate and consistent per person. Evaluations show that using IK on the results of a SOTA 3D HPE model to estimate the mesh pose combined with our A2B model's shape parameters leads to superior and consistent results compared to existing HME models. Moreover, HME models also benefit from our approach. Replacing their estimated shape parameters with the A2B shape parameters created from (pseudo) GT anthropometric measurements leads to an improvement of their score and consistent body shapes.

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
