# Leveraging Anthropometric Measurements to Improve Human Mesh Estimation and Ensure Consistent Body Shapes
# Supplementary Material

## A Anthropometric Measurements

The selection of the anthropometric measurements is mainly adopted from AnthroNet (Picetti et al., 2023). In total, 36 measurements are selected, which can be divided into 23 lengths and 13 circumferences. All measurements are taken based on the standard SMPL-X T-pose. The reference landmarks are chosen by matching the vertices on the default mesh with the landmarks defined by the anthropometric survey of the U.S. army personnel (Gordon et al., 2014). A visualization of the landmarks can be found in Figure 1 and 2a. The lengths are calculated by computing the Euclidean distance between two landmarks or the difference along the coordinate axis pointing upwards for certain heights. The lenghts are visualized in Figure 2b and 3. Table 1 lists the enclosing landmarks for each length. To measure the circumferences, we adopt the code from (Bojanic, 2023). For each measurement, a plane is created, the intersection between the mesh and the plane are extracted and the convex hull of the result is calculated. During this process, the mesh is restricted to the body part to be measured. A visualization of the circumferences can be found in Figure 4 and a list of the landmarks and the normal vectors spanning the plane in Table 2.

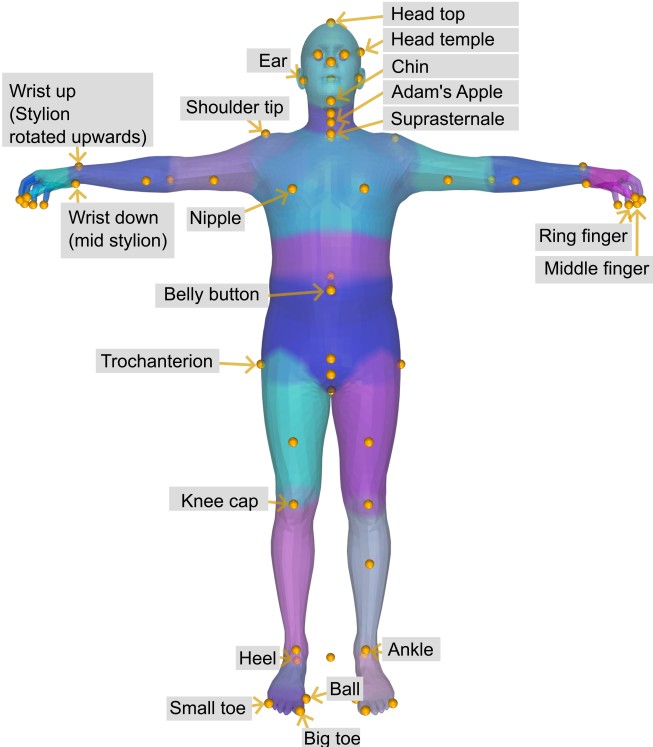

Figure 1: Visualization of the used landmarks with a standard T-pose SMPL-X mesh in front view.

Table 1: Definitions of lengths by their two enclosing landmarks.

| Idx | Length Name | From | To |
|---|---|---|---|
| 1 | Shoulder width | Left shoulder tip (left acromion) | Right shoulder tip |
| 2 | Back torso height | Cervicale | Back belly button |
| 3 | Front torso height | Suprasternale (top of the breast bone) | Belly button |
| 4 | Head | Head top | Cervicale |
| 5 | Midline neck | Chin | Suprasternale |
| 6 | Lateral neck | Center between the ears | Cervicale |
| 7 | Height | Head top | Center between heels |
| 8/9 | Hand right/left | Center between middle and ring finger | Stylion rotated downwards |
| 10/11 | Arm right/left | Acromion | Wrist |
| 12/13 | Forearm right/left | Elbow | Stylion rotated downwards |
| 14/15 | Thigh right/left | Outer point at the femur (Trochanterion) | Knee cap |
| 16/17 | Calf right/left | Knee cap | Ankle |
| 18/19 | Foot width right/left | Small toe | Big toe |
| 20/21 | Heel to ball right/left | Heel | Ball |
| 22/23 | Heel to toe right/left | Heel | Big toe |

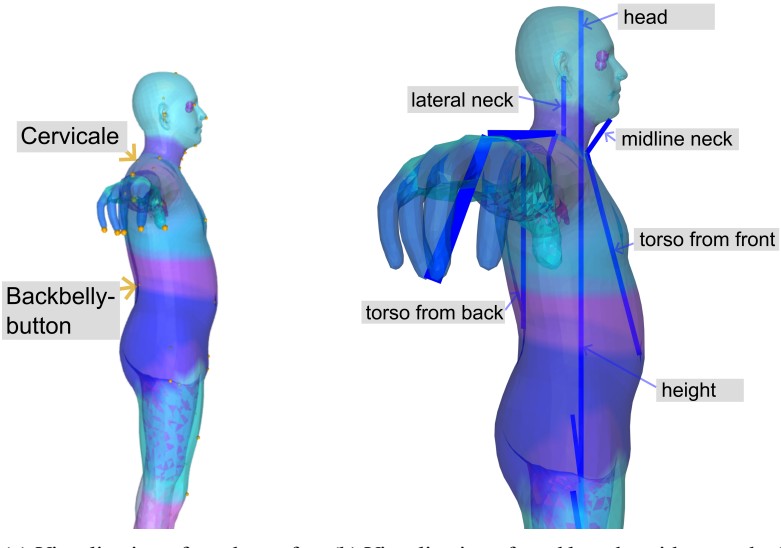

(a) Visualization of a subset of the used landmarks.

(b) Visualization of used lengths with a standard T-pose SMPL-X mesh.

Figure 2: Sideview visualizations of landmarks (a) and lengths (b).

Table 2: Definitions of circumferences by landmarks and the normal vector spanning the plane.

| Idx | Circumference | Normal Vector | Position |
|---|---|---|---|
| 1 | Waist | Up | Belly button |
| 2 | Chest | Up | Nipple |
| 3 | Hip | Up | Pubic bone |
| 4 | Head | Up | Head temple |
| 5 | Neck | Spine to head | Adam's apple |
| 6/7 | Upper Arm | Shoulder to elbow | Center of the bicep |
| 8/9 | Forearm | Elbow to wrist | Widest point of the forearm |
| 10/11 | Thigh | Up | Center of the thigh |
| 12/13 | Calf | Up | Widest point of the calf |

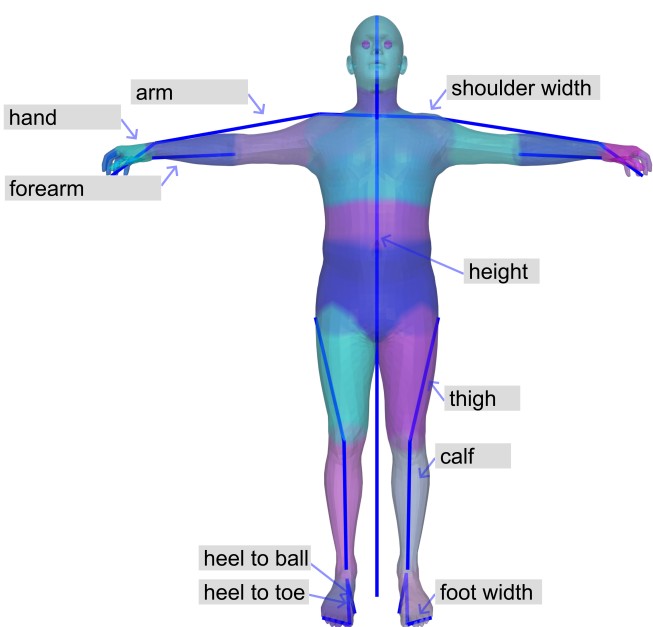

Figure 3: Visualization of used lengths with a standard T-pose SMPL-X mesh in front view.

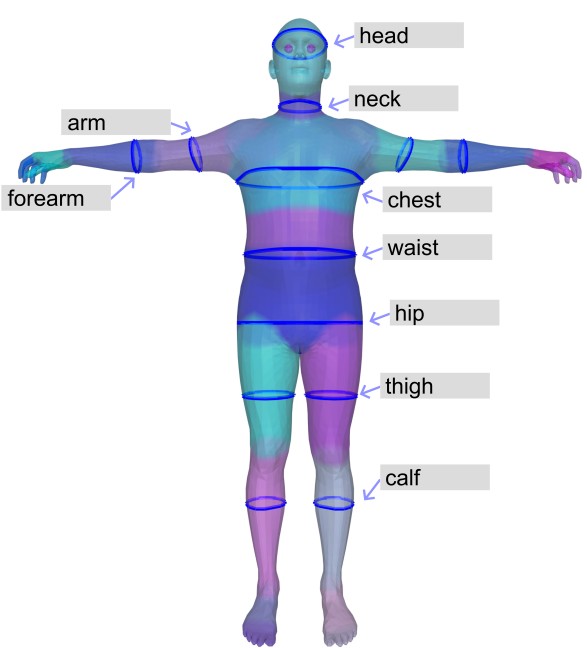

Figure 4: Visualization of used circumferences with a standard T-pose SMPL-X mesh in front view.

## B    3D HUMAN SHAPE GROUND TRUTH ANALYSIS

We further analyze the GT shape consistency for the common datasets Human3.6m (Ionescu et al., 2014) and MPI-INF-3DHP (Mehta et al., 2017). We find that for Human3.6m, the bone lengths derived from the 3D annotations are fixed, but not for MPI-INF-3DHP. Therefore, we do not report the deviations of 3D joint annotations for Human3.6m, since there are none. We further evaluate the SMPL-X annotations for both datasets provided by NeuralAnnot (Moon et al., 2022) which are used by HME models as GT for training. See Tables 3, 4 for details.

Table 3: GT data analysis for MPI-INF-3DHP (Mehta et al., 2017). Bone length analysis based on the 3D joint locations (left) and on SMPL-X annotations by NeuralAnnot (right). Standard deviation $\sigma$, relative standard deviation $\frac{\sigma}{avg}$ and relative range $\frac{max-min}{avg}$ of anthropometric measurements are reported. Standard deviations are given in cm, except for the $\beta$ parameters. The values are averaged between left and right body parts and between all persons in each dataset. The $\beta$ parameter standard deviation is averaged over all $\beta$ parameters.

| 3D joint annotations | | | | SMPL-X annotations | | | |
|---|---|---|---|---|---|---|---|
| Measure | $\sigma$ | r. $\sigma$ | r. range | Measure | $\sigma$ | r. $\sigma$ | r. range |
| head | 0.19 | 1.03% | 2.08% | head | 0.21 | 0.75% | 4.87% |
| hip width | 0.22 | 0.89% | 1.80% | hip circ. | 1.16 | 1.16% | 9.13 % |
| forearm | 0.21 | 0.87% | 1.77% | forearm | 0.45 | 1.80% | 9.75% |
| upper arm | 0.29 | 0.90% | 1.82% | arm | 0.83 | 1.59% | 8.19% |
| lower leg | 0.60 | 1.49% | 3.06% | lower leg | 1.05 | 2.56% | 11.54% |
| thigh | 3.83 | 7.91 % | 41.90% | thigh | 0.77 | 2.02% | 9.47% |
| | | | | height | 2.76 | 1.56% | 8.24% |
| | | | | $\beta$ param. | 0.18 | | |

Table 4: GT data analysis for Human3.6m (Mehta et al., 2017): Analysis of SMPL-X annotations by NeuralAnnot. Standard deviation $\sigma$, relative standard deviation $\frac{\sigma}{avg}$ and relative range $\frac{max-min}{avg}$ of anthropometric measurements are reported. Standard deviations are given in cm, except for the $\beta$ parameters. The values are averaged between left and right body parts and between all persons in each dataset. The $\beta$ parameter standard deviation is averaged over all $\beta$ parameters.

| SMPL-X annotations | | | |
|---|---|---|---|
| Measure | $\sigma$ | r. $\sigma$ | r. range |
| head | 0.41 | 1.51% | 10.28% |
| hip circ. | 1.24 | 1.19% | 8.90% |
| forearm | 0.83 | 3.30% | 27.93% |
| arm | 0.77 | 2.58% | 22.88% |
| lower leg | 0.43 | 1.18% | 12.20% |
| thigh | 0.66 | 1.27% | 9.43% |
| height | 3.40 | 2.06% | 15.66% |
| $\beta$ param. | 0.20 | | |

## C EVALUATING A2B MODELS

We measure two types of errors to evaluate the performance of our A2B models. The first type ($\beta$ error) shows the error if we take the GT $\beta$ parameters, derive anthropometric measurements (B2A), input them into the A2B models and evaluate the MSE of the predicted $\beta$ parameters. The second type ($A$ error) calculates B2A from the predicted $\beta$ parameters and evaluates the mean difference between the GT and predicted anthropometric measurements (all 36) in mm. These evaluations are a kind of cycle consistency evaluation for A2B and B2A. Figure 5 provides a visualization of the evaluation scheme. The part that is also included in the training is highlighted.

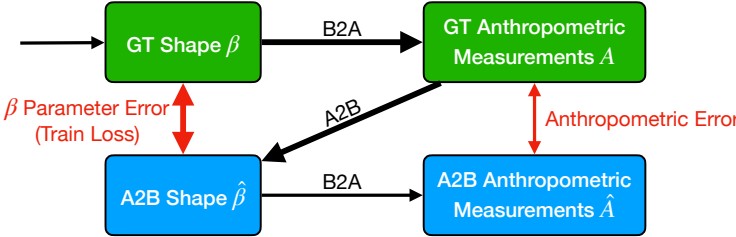

Figure 5: Visualization of the A2B evaluation and training procedures. The part involved in training the A2B models is highlighted with thicker arrows. B2A is a deterministic function and not learned.

## D    Keypoint Selection for fit3D

We use the fit3D (Fieraru et al., 2021) dataset for our evaluations, since this is the only sports dataset with public SMPL-X annotations. We evaluate on the SMPL-X joints, since these are trivial to obtain from SMPL-X meshes and there is no regressor available for the fit3D annotated 3D joints. SMPL-X has 144 defined joints. Since our focus is mainly on the body and not on the hands and face, we remove most of these joints. In the end, we select a subset of 37 SMPL-X joints: pelvis, left hip, right hip, spine1, left knee, right knee, spine2, left ankle, right ankle, spine3, left foot, right foot, neck, left collar, right collar, head, left shoulder, right shoulder, left elbow, right elbow, left wrist, right wrist, left index, left thumb, right index, right thumb, left big toe, left small toe, left heel, right big toe, right small toe, right heel, right eye, left eye, right ear, left ear, nose.

## E    Generation of Pseudo GT Anthropometric Measurements

As we do not have access to the athletes of ASPset and fit3d to obtain real anthropometric measurements, we need an alternative to kind-of simulate this process. For ASPset, as a first step, we run IK on the GT 3D joint locations. We obtain the necessary anthropometric parameters from the generated meshes with B2A. Then, we use the median values of these measurements as the GT anthropometric values. We call these parameters *pseudo GT* throughout this paper, since this is not directly the GT, but obtained from IK executed on the GT 3D joint locations and the B2A computation from the created meshes. These parameters are used in this paper to generate the pseudo GT $\beta$ parameters by A2B prediction.

We do not have access to the athletes of the fit3D dataset either. Therefore, we need some kind of GT data to mimic measurements. Obviously, there is no GT available for the official test set evaluation on the evaluation server. We therefore split the official training dataset into a training, validation, and test set for our evaluations. Details can be found in the main paper. With this selection, we have GT shape parameters available. We do not use these directly, but apply B2A and use the median measurements over time in order to mimic the measuring process and obtain a single set of anthropometric measurements per person (which is not the case for the provided GT parameters, see Section 3 in the main paper). In real applications, this step is omitted because the anthropometric parameters can be measured directly from the athletes before starting the recording.

## F    Finetuning HME Models with Pseudo GT Meshes

Finetuning existing HME models on pure 3D joints datasets is not possible, since they need mesh annotations for training. However, with IK, we can generate pseudo GT meshes. We exemplary test a finetuning of SMPLer-X on ASPset with this approach. Experiments show that using their finetuning script with 1.6M iterations leads to worse results than the results without finetuning. Therefore, we reduce the number of iterations with early stopping and achieve better results with finetuning only for 32K iterations.

The results shown in Table 5 prove that finetuning on IK generated meshes can lead to a significant improvement of the scores. Replacing the $\beta$ parameters of the finetuned results with the A2B $\beta$ parameters boosts the performance even more. These are the best results achieved with any existing HME model throughout this study.

Moreover, we experiment with using the SMPLer-X body shape parameters combined with the poses estimated by IK applied to the UU results (see last two rows of Table 5). Using the $\beta$ parameters from SMPLer-X leads to a slightly better result than the original 3D joint based result (without IK). This evaluation shows that 3D HPE models are better in precisely locating the joints of humans than HME models, but HME models are better in estimating the shape of humans. We also try to use the $\beta$ parameters of the finetuned variant together with the UU IK poses like before. However, this resulted in a performance drop compared to the body shape parameters from the original SMPLer-X without finetuning. These experiments show that finetuning HME models on pseudo ground truth leads to a better performance regarding the keypoints, but the estimated $\beta$ parameters have worse quality. This can further be proven by replacing the $\beta$ parameters from the finetuned SMPLer-X variant with the $\beta$ parameters from the not finetuned model, which results in a performance gain of over 5 mm compared to the original results from the finetuned version (rows 2 and 4 in Tab.

Table 5: MPJPE results in mm for the test split of ASPset. Results are given for different methods and replaced $beta$ parameters with A2B results (columns NN/SVR) or the median of the original $\beta$ parameters from the model noted in the *measurements* column. SMPLer-X FT stands for the best finetuned variant of SMPLer-X (finetuned with the meshes obtained from IK executed on the GT 3D joints). The *orig* column contains the results without replaced $\beta$ parameters. We highlight the best result for each model and the best option for the combination of UU IK pose and SMPLer-X $\beta$ parameters, since this combination outperforms the original UU IK result, too.

| model | orig. | measurements | NN m | SVR m | NN n | SVR n | median |
|---|---|---|---|---|---|---|---|
| SMPLer-X | 86.02 | SMPLer-X | 85.89 | 85.69 | 86.03 | 85.99 | 86.04 |
| SMPLer-X FT | 79.09 | SMPLer-X FT | 78.92 | 78.88 | 79.44 | 79.37 | 79.44 |
| SMPLer-X FT | - | GT | 65.63 | 65.84 | **64.71** | 64.76 | - |
| SMPLer-X FT | - | SMPLer-X | 73.41 | 73.29 | 73.65 | 73.63 | 73.66 |
| UU IK | 67.54 | UU | 66.92 | 66.60 | 67.25 | 67.12 | 67.16 |
| UU IK | - | SMPLer-X | 63.80 | 63.64 | 63.81 | 63.78 | 63.82 |
| UU IK | - | SMPLer-X FT | 69.46 | 69.27 | 69.70 | 69.63 | 69.69 |
| UU IK | - | GT | 56.44 | 56.56 | **55.14** | 55.19 | - |

5). However, our method using the UU IK poses and the A2B body shape parameters with GT anthropometric measurments achieves the overall best results.

We provide a comprehensive summary and visualization of all results on the ASPset dataset in Section G. This includes results of existing HME models, results of our approach, and the finetuning results.

# G SUMMARY OF THE RESULTS

We execute a multitude of experiments with different combinations of pose and shape parameters. Figure 7 summarizes the results with their pose and shape origins for ASPset. In general, the poses estimated by IK based on the UU results (red branch in Fig. 7) are more precise than the poses estimated by SMPLer-X (light blue branch in Fig. 7). Further, the body shape parameters from our A2B models with GT anthropometric measurements (green boxes in Fig. 7) achieve the best results for all poses. We provide more qualitative examples comparing SMPler-X with this approach in

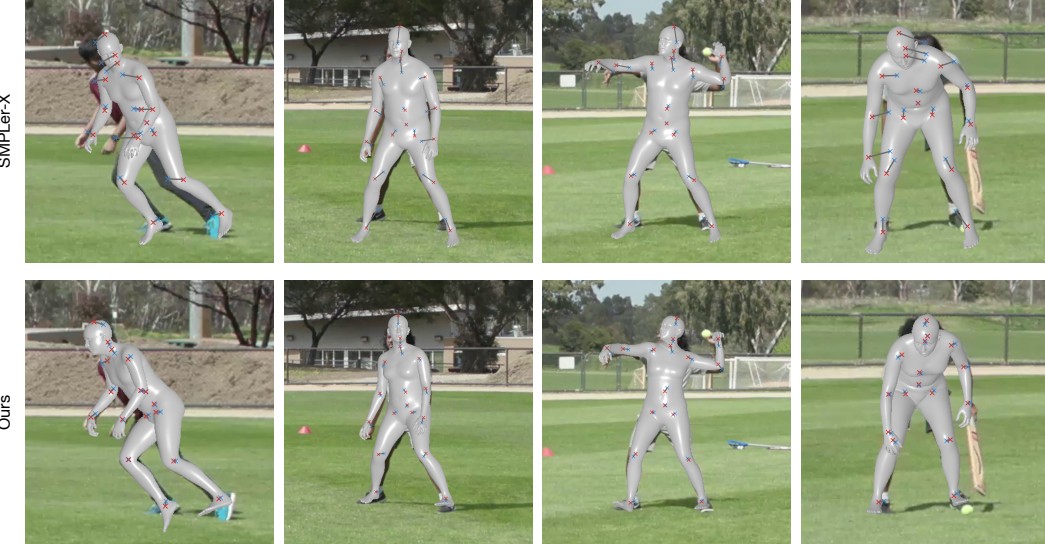

Figure 6: Qualitative results of SMPLer-X and our approach for example frames from ASPSet. GT joints and estimated joints are color-coded. Corresponding joints are connected.

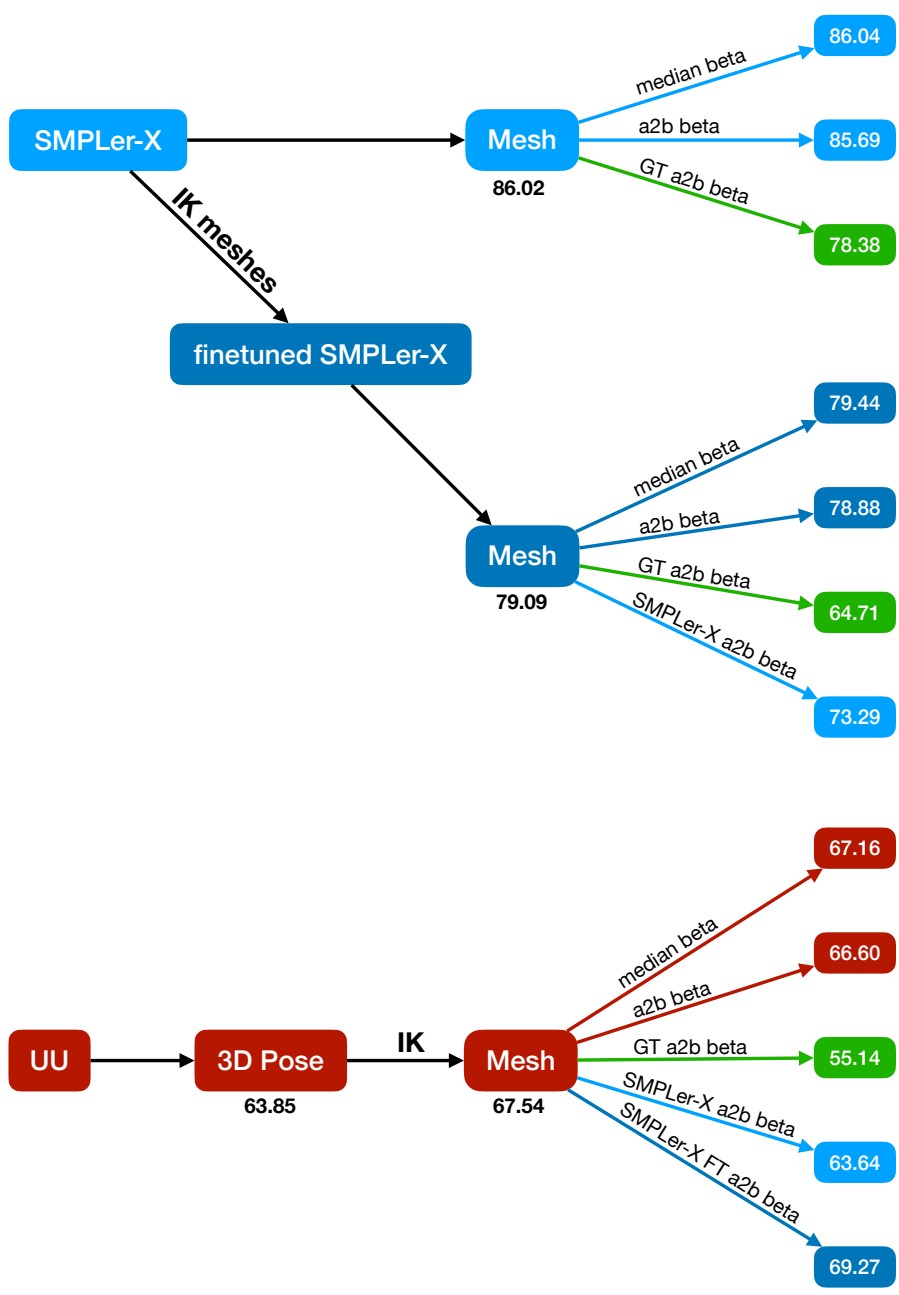

Figure 7: Overview of the main results for the ASPset dataset. All results are MPJPE results in mm. Results below *mesh* boxes show the result with the original $\beta$ parameters. All results after arrows to the right are results with replaced $\beta$ parameters. The type of the $\beta$ parameters is noted on the arrow and is color-coded: pseudo GT (green), SMPler-X (light blue), SMPler-X finetuned (dark blue), UU IK (red).

Figure 6. Without access to the GT, all models benefit slightly from A2B model results with the median anthropometric measurements from B2A of the estimated meshes by the respective model (boxes with same color for all three branches in Fig. 7). Moreover, SMPLer-X A2B body shape parameters perform best when analyzing body shapes without GT access (light blue boxes in Fig. 7). Finetuning SMPLer-X with IK created meshes (dark blue branch in Fig. 7) improves the performance of SMPLer-X, although the quality of the body shape deteriorates. This can be seen as by comparing the shapes from SMPLer-X and finetuned SMPLer-X (dark blue and light blue boxes in Fig. 7) with finetuned and IK poses.

Since fit3D is a larger dataset, finetuning UU works better, which further leads to better IK meshes with an MPJPE of 37.02 mm. Enforcing consistent meshes with GT or IK A2B shape parameters decreases the performance slightly in this case. However, A2B shape parameters achieve slightly better scores than median values. This also holds for OSX and Multi-HMR. Overall, the approach with UU, IK, and A2B body shape parameters achieves an over 33 mm lower MPJPE than any HME model.