# OpenReview forum: "Leveraging Anthropometric Measurements to Improve Human Mesh Estimation and Ensure Consistent Body Shapes"
_ICLR.cc/2025/Conference — ICLR 2025 Conference Withdrawn Submission_

### Official Review · Reviewer_2yYB · 2024-10-16

**Soundness:** 2
**Presentation:** 2
**Contribution:** 2
**Rating:** 3
**Confidence:** 4

**Summary:**

This paper presents a 3D human mesh estimation system that utilizes human anthropometric information for more accurate 3D body shape. To this end, the authors first introduce the A2B model, a model that converts human anthropometric information to the SMPL-X shape parameter. Based on the A2B model, the authors report two cases where the A2B model could be useful. First, while keeping estimated 3D pose from prior 3D human mesh estimation methods and changing the shape parameter to theirs, 3D errors become less. Second, from 2D and 3D human keypoint estimators, the authors perform the inverse kinematics using the shape parameter from the A2B model.

**Strengths:**

As the authors mentioned, most works are focusing only on estimating correct 3D poses without careful consideration of the 3D body shape. The pursuing research direction of this paper is interesting.

**Weaknesses:**

1. Given anthropometric information, regressing 3D body shape parameters is quite straightforward as we could get a large amount of synthetic datasets to train the regressor like the authors did. The problem is that given a single image/video, getting the accurate anthropometric information is not trivial. The authors said that they can use real anthropometric information if they are available in real-world applications (Fig. 3). In the real world, getting such anthropometric information is not trivial as we hardly bring measurements in our daily life. Instead, the authors used GT or median of predicted shape parameters from off-the-shelf estimators if the video of a single person is available. First, using GT does not make sense at all. Assuming GT information during the test is cheating. Also, using median value and converting it with B2A and A2B models (L372) do not make sense at all. This is kind of averaging shape parameters from previous works, not predicting actual 3D body shape based on the image evidence. Hence, I do not think the effectiveness and benefit of the proposed A2B model is justified.

2. The two use cases that the authors introduced are not impressive. First, changing the shape parameter to the A2B’s output uses GT (L347). This is cheating and makes the comparison unfair. Also, changing the shape parameter could make the results worse as it changes the bone length as well. With the same 3D pose and changed bone length, the 3D positions of keypoints could change, which could make 3D errors higher. Second, performing the inverse kinematics based on the off-the-shelf estimators’ output is quite widely used and is not novel at all. Actually, many people try to avoid this when designing 3D pose estimators as it makes the running time greatly slow due to the iterative optimization. Although the authors tried to make it fast (L460), I’m guessing the running time is slower than 5 fps.

3. There are no qualitative results that show the body shape becomes better. There are few qualitative results, but most of them are trying to show corrected 3D pose after the inverse kinematics. As mentioned in the above, the inverse kinematics is quite straightforward and getting better at the cost of the running time is expected. Instead, there should be side-by-side comparison that show the proposed one is able to recover accurate body shapes for diverse humans, for example,fat/thin persons.

**Questions:**

N/A

---

### Official Review · Reviewer_Jbmk · 2024-10-29

**Soundness:** 3
**Presentation:** 4
**Contribution:** 3
**Rating:** 5
**Confidence:** 4

**Summary:**

To address the issue of inconsistent shape parameters estimation, the authors used anthropometric measurements.

By replacing the originally estimated shape parameters with those obtained through the A2B model, they achieved performance improvements.

**Strengths:**

1. The approach of mapping complex shape parameters to more understandable anthropometric measurements is innovative.

2. The analysis of existing data to identify flaws in the original data is clearly explained and easy to understand.

3. The authors tried to demonstrate the superiority of the proposed method through various experiments.

4. The paper is well-written and easy to read.

**Weaknesses:**

1. Using anthropometric measurements obtained from GT in experiments does not appear fair (Table 3).

2. The experiment using the median of measurements as A2B input seems unjustified (Table 4).

3. Both experiments appear inadequate in addressing the issue of noise in the shape parameter estimates of the existing model.

4. Although the authors mention that there is no good means to correct the inaccurate GT at present, relying on noisy GT shape parameters to measure the performance may not sufficiently demonstrate the effectiveness of the proposed method.

**Questions:**

Are there datasets available that include anthropometric measurements GT? It would be helpful to see qualitative comparisons, such as comparing the output mesh from the existing method and that from the proposed method using the real anthropometric measurements of the authors or other person.

---

### Official Review · Reviewer_mHRQ · 2024-11-03

**Soundness:** 2
**Presentation:** 4
**Contribution:** 3
**Rating:** 5
**Confidence:** 3

**Summary:**

This paper propose a framework to retrieve shape-consistent human mesh estimation results using anthropometric measurements. The authors fit a small model (A2B model) that converts anthropometric measurements to SMPL-X shape parameters. The authors replace the shape estimation outputs from existing methods with the shape derived from the anthropometric measurements (from GT or IK) with their proposed A2B model, and observe improved MPJPE results on the test set. The authors point out that the datasets for evaluation themselves have inconsistent anthropometric measurements or shape parameters.

**Strengths:**

A substantive assessment of the strengths of the paper, touching on each of the following dimensions: originality, quality, clarity, and significance. We encourage reviewers to be broad in their definitions of originality and significance. For example, originality may arise from a new definition or problem formulation, creative combinations of existing ideas, application to a new domain, or removing limitations from prior results. You can incorporate Markdown and Latex into your review. See https://openreview.net/faq.
1. This paper is clearly written and easy to understand.
2. This paper propose a framework to improve shape consistent for HMR using anthropometric measurements.
3. The authors propose a model (A2B model) to convert anthropometric measurements to SMPL-X shape parameters, as well as training data generation methods for the model. The proposed model has values in fast registering new human subjects to the SMPL-X body model.
4. The authors provide the code and model along with their paper.

**Weaknesses:**

1. This paper does not provide sufficient explanation for how inconsistencies between anthropometric measurements and SMPL-X shape parameters lead to inconsistencies in human shapes. For example, there is no information on whether real subject anthropometric measurements  will remain unchanged under different poses or in motion (though obtaining measurements in the latter situation is challenging).
2. In evaluation, the authors use anthropometric measurements derived from GT subject shapes (``pseudo GT'' in the text). Considering the A2B model has low error, this could be equivalent to feeding actual GT shapes during evaluation. However, there is no direct comparison between results using converted shapes from pseudo GT and those using actual GT shapes.
3. The only metric used in evaluation is MPJPE. The authors could evaluate their predicted results using the same criteria applied when they analyze the shape inconsistencies within existing datasets.

**Questions:**

See the weakness above.  If the authors can fully address the weaknesses above, I am willing to re-consider my scores.

---

### Official Review · Reviewer_S1Db · 2024-11-03

**Soundness:** 3
**Presentation:** 3
**Contribution:** 3
**Rating:** 6
**Confidence:** 4

**Summary:**

This paper proposes an intuitive way to improve the estimation of human body shape: learn a mapping model from easily observed spatial measurements, here using anthropometric measurements, to the PCA parameter betas of the SMPL model.

This paper conducts a thorough analysis of the ASPset and fit3D dataset. It proves the inconsistency in temporal distribution, which should be blamed on the SMPL model's imperfect decoupling of theta and betas parameters.

**Strengths:**

The perspective of the paper is clear and useful. The method is quite simple and intuitive, by learning a mapping from anthropometric measurements to the betas parameter. The anthropometric measurements include 23 lengths and 13 circumferences. The author trained their method on the Agora dataset, which has the existing largest individual human shapes.

The A2B method could also help Other SOTA HME methods. The experiments could prove this.

The complete pipeline that combines HPE, A2B, and IK models aligns well with the core idea that we solve body shape from explicit spatial clues and use the pre-trained A2B model which possesses the body shape prior knowledge to solve the body shape.

**Weaknesses:**

Not enough quality results are shown to prove the idea.

The paper focuses too much on expressing the operation details but has not shown enough vivid digrams.

I am confused about how to get the anthropometric measurements on some random videos. The HPE model could only help to get the bone length, but how to get the extra 13 circumferences? This part is not clear.

**Questions:**

Please help me solve the question of how to get the extra 13 circumferences. If you could do this on any given videos, why do you only compare on such few datasets? The experiments on popular video test benchmarks are not included, like emdb, human3.6m, 3dpw, etc.

---

### Note · Authors · 2024-11-14

I have read and agree with the venue's withdrawal policy on behalf of myself and my co-authors.